# We shape our buildings, but do they then shape us? A longitudinal analysis of pedestrian flows and development activity in Melbourne

**Andres Sevtsuk** *, **Rounaq Basu, Bahij Chancey**

Department of Urban Studies and Planning, Massachusetts Institute of Technology, Cambridge, MA, United States of America

* asevtsuk@mit.edu

**Data Availability Statement:** We have uploaded the dataset as a CSV file with this submission.

**Funding:** The author(s) received no specific funding for this work.

## Abstract

Cities are increasingly promoting walkability to tackle climate change, improve urban quality of life, and address socioeconomic inequities that auto-oriented development tends to exacerbate, prompting a need for predictive pedestrian flow models. This paper implements a novel network-based pedestrian flow model at a property-level resolution in the City of Melbourne. Data on Melbourne's urban form, land-uses, amenities, and pedestrian walkways as well as weather conditions are used to predict pedestrian flows between different land-use pairs, which are subsequently calibrated against hourly observed pedestrian counts from automated sensors. Calibration allows the model extrapolate pedestrian flows on all streets throughout the city center based on reliable baseline observations, and to forecast how new development projects will change existing pedestrian flows. Longitudinal data availability also allows us to validate how accurate such predictions are by comparing model results to actual pedestrian counts observed in following years. Updating the built-environment data annually, we (1) test the accuracy of different calibration techniques for predicting foot-traffic on the city's streets in subsequent years; (2) assess how changes in the built environment affect changes in foot-traffic; (3) analyze which pedestrian origin-destination flows explain observed foot-traffic during three peak weekday periods; and (4) assess the stability of model predictions over time. We find that annual changes in the built environment have a significant and measurable impact on the spatial distribution of Melbourne's pedestrian flows. We hope this novel framework can be used by planners to implement "pedestrian impact assessments" for newly planned developments, which can complement traditional vehicular "traffic impact assessments".

## Introduction

After a century of automobile-oriented urban growth, many city governments around the world are implementing extensive pedestrian improvements on their streets, banking on a

**Competing interests:** The authors have declared
that no competing interests exist.

future with less space dedicated to cars and more to activities that take place on foot. Walkable
cities reduce urban energy consumption [1–4], enhance public health [5–7], build community,
and foster social cohesion [8–10]. Dense, diverse, and walkable city centers have also become a
magnet for global talent, who work in knowledge-intensive jobs and cherish serendipitous
face-to-face encounters that flourish in dense and liveable city environments [11, 12]. Even in
the American context, where large cities are more suburban than their international counter-
parts, inner-city areas have witnessed both residential and job growth in the past decade [13,
14], pressuring city governments to respond with development plans that foster walkability.

In order to forecast the effects that proposed built-environment changes could have on foot
traffic, cities increasingly require quantitative models [15–19]. New real estate and infrastruc-
ture developments will not only fit into existing mobility patterns, but will themselves also
influence travel behavior [20]. Some developments will decrease walking activity, while others
will increase it. Yet transportation models have historically centered largely on vehicular traffic
[21, 22]. Travel demand models, if they include pedestrian trips at all, focus on aggregate
pedestrian trip volumes, not their flows along city streets. However, pedestrian flows are not
evenly spread throughout neighborhoods, but rather concentrated on certain streets. In order
for cities to enhance walking, it is important to not only estimate how many total walking trips
a planned area might generate, but also how these trips are likely to distribute along specific
streets [23]. This need has become especially apparent in the context of COVID-19, where
many cities have introduced pilot programs to expand pedestrian space in the form of "shared
streets" programs [24, 25], but failed to attract expected levels of use, in part due to selection of
pilot locations that do not witness much pedestrian use [26]. A renewed policy emphasis on
better pedestrian environments has thus triggered a need to better understand how pedestrian
flows are shaped by the built environment in general, and how building and infrastructure
developments might affect foot-traffic on surrounding streets in particular.

This paper applies a recently developed pedestrian flow model [10, 41] to examine the lon-
gitudinal impacts of changes in the built environment on pedestrian activity for the first time.
As opposed to aggregate estimations of pedestrian activity with respect to broad neighborhood
characteristics such as the three (or more) neighborhood D's—density, diversity and design
[27]—which much of urban walkability literature has centered on [28–31], our model relies on
an address-level analysis of pedestrian accessibility and destination choice, combining trip-
generation, trip-distribution, and route-choice into an accessibility-driven framework that
some planning and mobility scholars have called for [32].

We showcase an application of the model in Melbourne, Australia—often rated as one of
the most livable cities in the world [33]. Besides making walkability a public priority, Mel-
bourne also stands out for its systematic data collection of pedestrian flows on city streets,
which is used to inform urban development and transportation policy decisions [34, 35]. The
availability of automated pedestrian count data over several years provides a unique advantage
to applying the model in Melbourne—not only do the data offer a reliable benchmark to cali-
brate a predictive model, the longitudinal nature of the data also makes it possible to validate
the accuracy of model predictions over time. We use Melbourne's pedestrian count panel data
together with known weather conditions at each observation period to calibrate a network
model of pedestrian flows and explore how the calibrated model can be used to forecast
changes in foot-traffic due to new developments that shift the location and size-balance of resi-
dents, jobs, and business establishments in the city each year. We develop a separate pedestrian
flow model for three representative workday periods—the morning rush hour (8:00–9:00), the
lunch hour (13:00–14:00), and the evening rush hour (17:00–18:00)—that are used to predict
changes in foot-traffic for each of the five subsequent years based on annually updated built
environment data. In addition to examining changes in flows attributable to development

activity, we also examine whether the relationships between pedestrian flows and the built environment captured in model coefficients remains stable over time, or whether pedestrians' behavioral responses to the built environment change over time due to shifting preferences and socio-cultural evolution [36]. The key technical innovations of the paper lie in (1) using a much more precise, longitudinal dataset of pedestrian counts from automated sensors as dependent variables than available in prior research; (2) developing predictive models where the accuracy of the predictions can be validated using observed data in the following years; and (3) using machine learning estimation techniques to tackle issues of multi-collinearity and non-linear relationships in the data.

Our findings suggest that pedestrian flows are strongly shaped by the structure of the built environment. Changes in residential, job, and amenity locations produce predictable changes in pedestrian flows during different times of the day. Longitudinal assessments of predictive accuracy suggest such that once calibrated, model coefficients are most reliable for the first year of prediction, and remain fairly stable (albeit with some loss in accuracy) during the following four years. The proposed model thus demonstrates the feasibility of "pedestrian impact assessments", which can inform how new development activity in dense neighborhoods could affect pedestrian flow dynamics over time.

## Materials and methods

### Data

Melbourne's Census of Land Use and Employment (CLUE) summarizes the state of the built environment in the city annually by describing the spatial distribution of land uses, employment, and economic activity at the property level resolution [37]. CLUE data include the number of residential units by property, business establishments and industry codes by property, the number of restaurant seats by establishment, and total employment by block. Fig 1 shows the spatial extent of the study area and Table 1 describes changes in the Melbourne's built environment between 2013–2018 according to CLUE records. On average, around 630,000 square meters of new space is constructed in our study area each year.

Additional data from the City of Melbourne included the pedestrian network (we use outdoor sidewalks and crosswalks only, with separate walkways on either side of the roadway, zebras, signalized crossings etc.), locations and capacities of commercial parking lots, locations and areas of parks, and enrollment numbers for local universities. State of Victoria data also included the location of tram and train stations and the number of lines serving each station, which we used as weights. The Royal Melbourne Institute of Technology and the University of Melbourne facilities information provided the location of campus buildings. Our models merge these datasets to generate weighted origin-destination flows between individual buildings, amenities, stations, parks, parking lots, and tourist attractions. We also used weather information on daily precipitation from State of Victoria and half-hour temperature measurements from the United States National Oceanic and Atmospheric Administration (US-NOAA) to fit our models.

We obtained pedestrian count data from Melbourne's automated Pedestrian Counting System [62]. The descriptive statistics of the 32 sensors that were consistently active from 2014–2018 during AM, lunch, and PM periods, illustrated in Table 2, suggest that overall foot-traffic in the city has been growing year on year, with the biggest increase between 2014–15. This is consistent with the built environment data in Table 1, which also suggested largest growth between 2014–2015 when the number of residential units in our study area increased by 13%. With respect to the three analysis periods, the smallest changes over the years are seen during the lunch period in Table 2—we suspect that morning and evening flows may be more affected by shifting

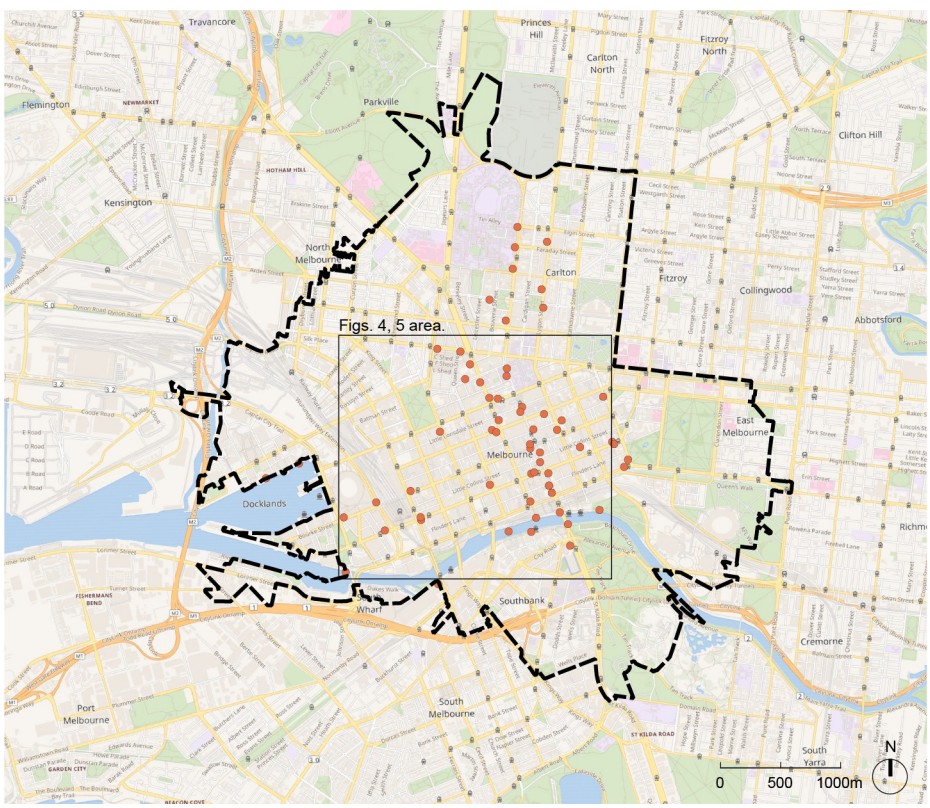

**Fig 1. Study area (thick dashed line) context map in Greater Melbourne.** Orange dots indicate sensor locations. Inset rectangle delineates the area covered in Figs 4 and 5. Wikimedia base map accessed under Open Data Commons Open Database License (ODbL), created by OpenStreetMap contributors.

transportation policies and behavior over the years (e.g. public transit fare pricing, parking policies, flexible starting hour), while lunch-time trips tend to be less impacted by such shifts. As expected, pedestrian counts are slightly lower on rainy days than dry days (S1 Fig in S1 File).

**Table 1. Change in built environment in Melbourne CBD, 2014–2018.** (Source: City of Melbourne 2021).

| year | | residential units | jobs | firms | amenities | food/beverage dining seats | total GFA (m²) | New GFA (m²) |
|------|---|-----|-----|-----|-----|-----|-----|-----|
| 2013 | # | 42,716 | 461,171 | 13,177 | 6,714 | 162,269 | 15,938,561 | |
| 2014 | # | 45,677 | 469,358 | 13,205 | 6,671 | 170,660 | 16,588,142 | 649,581 |
| | Δ | 6.9% | 1.8% | 0.2% | -0.6% | 5.2% | 4.1% | |
| 2015 | # | 51,761 | 468,370 | 13,196 | 6,643 | 173,390 | 17,310,027 | 721,885 |
| | Δ | 13.3% | -0.2% | -0.1% | -0.4% | 1.6% | 4.4% | |
| 2016 | # | 54,513 | 478,182 | 13,539 | 6,704 | 177,862 | 18,038,180 | 728,153 |
| | Δ | 5.3% | 2.1% | 2.6% | 0.9% | 2.6% | 4.2% | |
| 2017 | # | 59,570 | 485,457 | 13,727 | 6,744 | 179,718 | 18,515,413 | 477,233 |
| | Δ | 9.3% | 1.5% | 1.4% | 0.6% | 1.0% | 2.6% | |
| 2018 | # | 65,228 | 503,530 | 13,758 | 6,705 | 176,620 | 19,127,137 | 611,724 |
| | Δ | 9.5% | 3.7% | 0.2% | -0.6% | -1.7% | 3.3% | |

N = 7,790 property lots in study area

Δ is change over preceding year

**Table 2. Descriptive statistics of June weekday pedestrian counts from 32 sensors that were consistently active from 2014–2018 during AM, lunch, and PM periods.** (Source: City of Melbourne 2020).

|  | 2014 | 2015 | 2016 | 2017 | 2018 |
|---|---|---|---|---|---|
| **AM** |  |  |  |  |  |
| Mean | 889.1 | 991.9 | 1029.3 | 1080.2 | 1115.4 |
| Median | 607.5 | 644.5 | 642.5 | 697 | 685.5 |
| Min. | 107 | 125 | 84 | 59 | 130 |
| Max. | 3086 | 3789 | 3887 | 4146 | 4146 |
| Δ in mean |  | 11.6% | 3.8% | 4.9% | 3.3% |
| **Lunch** |  |  |  |  |  |
| Mean | 1157.2 | 1285.3 | 1311.6 | 1329.2 | 1357.2 |
| Median | 810.5 | 905 | 938 | 972 | 1047.5 |
| Min. | 177 | 217 | 188 | 142 | 234 |
| Max. | 3620 | 3716 | 3616 | 3512 | 3563 |
| Δ in mean |  | 11.1% | 2.0% | 1.3% | 2.1% |
| **PM** |  |  |  |  |  |
| Mean | 1304.8 | 1487.5 | 1549.3 | 1630.1 | 1709.8 |
| Median | 955 | 1070 | 1059.5 | 1118.5 | 1191 |
| Min. | 144 | 182 | 163 | 101 | 185 |
| Max. | 3592 | 4456 | 4539 | 4846 | 4749 |
| Δ in mean |  | 14.0% | 4.2% | 5.2% | 4.9% |

## Modeling pedestrian flows

Pedestrian flows between individual origin-destination (O-D) pairs in the city are modeled using a modified betweenness algorithm [38], which has been adapted to a number of different transportation applications [10, 15, 19, 39, 40]. The betweenness of a street segment is conventionally defined as the number of walks between origin-destination pairs in a network that pass through that segment. Given a set of trip origins and destinations, the index colloquially captures the flow—or in this case the foot-traffic—that is estimated to pass through each street segment.

In this study, we implemented the betweenness algorithm featured in the Urban Network Analysis toolbox [34], which includes six adjustments that make the analysis suitable to pedestrian flows [10, 41]. First, instead of modeling interactions between all O-D pairs, as is common in social network analyses [42], users can choose which O-D locations to include. This enables the model to capture trips from workplaces to transit stations or homes to parks, for instance. Second, trip lengths are limited by a "search radius" variable. This ensures that walking trips are assigned to only those destinations (e.g. transit stations) that are within reasonable distance [43]. Third, the volume of trips starting at each origin is determined by a numeric attribute attached to the origin. The number of trips generated from a given address location thus depends on the number of residents, jobs or establishments at the said address. Fourth, the volume of trips allocated to destinations is also controlled by an accessibility function with distance decay, which ensures that closer destinations and more attractive destinations receive more trips than less accessible destinations. This makes the model somewhat elastic to destination availability, whereby origin locations that are surrounded by larger or closer destinations not only route proportionately more trips to such destinations, but also generate a higher rate of trips altogether [44] (S1 Note in S1 File). Fifth, the model does not assume pedestrians only take shortest available walking paths to destinations, but rather allocates an equal probability to each available walking path that is up to a given percentage longer than the shortest route. A

detour ratio of "1.15", for instance, assigns trips to a given destination along all available routes up to 15% longer than the shortest route. This offers a simplified approach for capturing otherwise complex and idiosyncratic route preferences among different individuals. Pedestrian route choice literature has shown that route attributes, such as safety, comfort or stimulation (e.g., historic quality), can significantly affect route choice [30, 45–50], but such preferences tend to vary across demographic groups and types of trips (e.g., walking to a park versus walking to work). There has been a debate whether pedestrian navigation is primarily driven by distance minimization or angular turn minimization [51–53], though in transportation literature, distance or travel time are commonly found to be the primary transportation cost attributes, with turns along the route, as well as other route attributes (e.g., businesses passed, vegetation, etc.) playing an important role as well [49, 50, 54]. The distance-based route probability assignment we use here has been shown to achieve a relatively good fit for predicting pedestrian flows [55]. These five adjustments produce the modified betweenness index for modeling pedestrian journeys:

$$Betweenness[i]^{r,dr} = \Sigma_{j,k \in G - \{i\}, d[j,k] \leq r \cdot dr)} \frac{n_{j,k}[i]}{n_{j,k}} \cdot W[j] \cdot \frac{W[k]^{\alpha}}{e^{\beta \cdot d[j,k]}} \tag{1}$$

where $Betweenness[i]^{r,dr}$ describes how many walks from surrounding origins [j] to destinations [k] in a network $G$ pass through location [i], such that the maximum distance between [j] and [k] is less than or equal to search radius $r$ multiplied by the detour ratio $dr$. If weights $W$ are attached to origin points [j], then the number of walks routed from [j] to [k] correspond to $W$, adjusted by the distance decay parameter $\beta$. $W[k]^{\alpha}$ represents the attractiveness of destination $k$, where the effect of its weight $W$ on trip generation is controlled by an exponent $\alpha$.

The index is visualized for a single origin-destination pair in Fig 2, where walks from an origin point with a weight of "100" are routed to a destination along all routes that are up to 15% longer than the shortest available route. Because the origin is 800 meters from the destination, the distance decay parameter $\beta = 0.002$ reduces the number of trips allocated to the destination from 100 to only 20.2 (100 / $e^{0.002 \times 800}$ = 20.2). Segments where the origin and destination locations meet the network necessarily have the maximum betweenness value of 20.2, with lower values on intermediary streets, depending on how many of the available 56 routes traverse them.

In situations, where origin points have multiple competing destinations available for the same trip purpose, our algorithm uses the Huff destination probability model [56] (S1 Note in S1 File). If, for instance, an employment location has numerous transit stations within a walking-range around it, then instead of routing a single trip to the nearest station, or alternatively a repetitive trip to each of the available stations, we assign trip probabilities based on gravity accessibilities to competing destinations available within the search radius [57]. In the data we use for Melbourne, each origin can have hundreds of competing destinations and we route a fraction of trips to each, based on their accessibility. The resulting "patronage betweenness" index offers a unique way of estimating and predicting the spatial distribution of pedestrian trips in built environments at a high resolution [33].

Appropriate pedestrian flow types, their corresponding O-D pairs, and assumed distances and detours that capture most of the observed variation in pedestrian flow during a particular period can vary in different contexts and land-use patterns. In order to select appropriate O-D flows for Melbourne's AM, lunch and PM peak periods, we identified nine potential attractors for pedestrian trips, shown in Fig 3, which constitute 36 possible origin-destination pairs for the model. Even though some pedestrian journeys might occur between each of them, we expected certain flows to dominate foot-traffic during peak travel times. We checked available

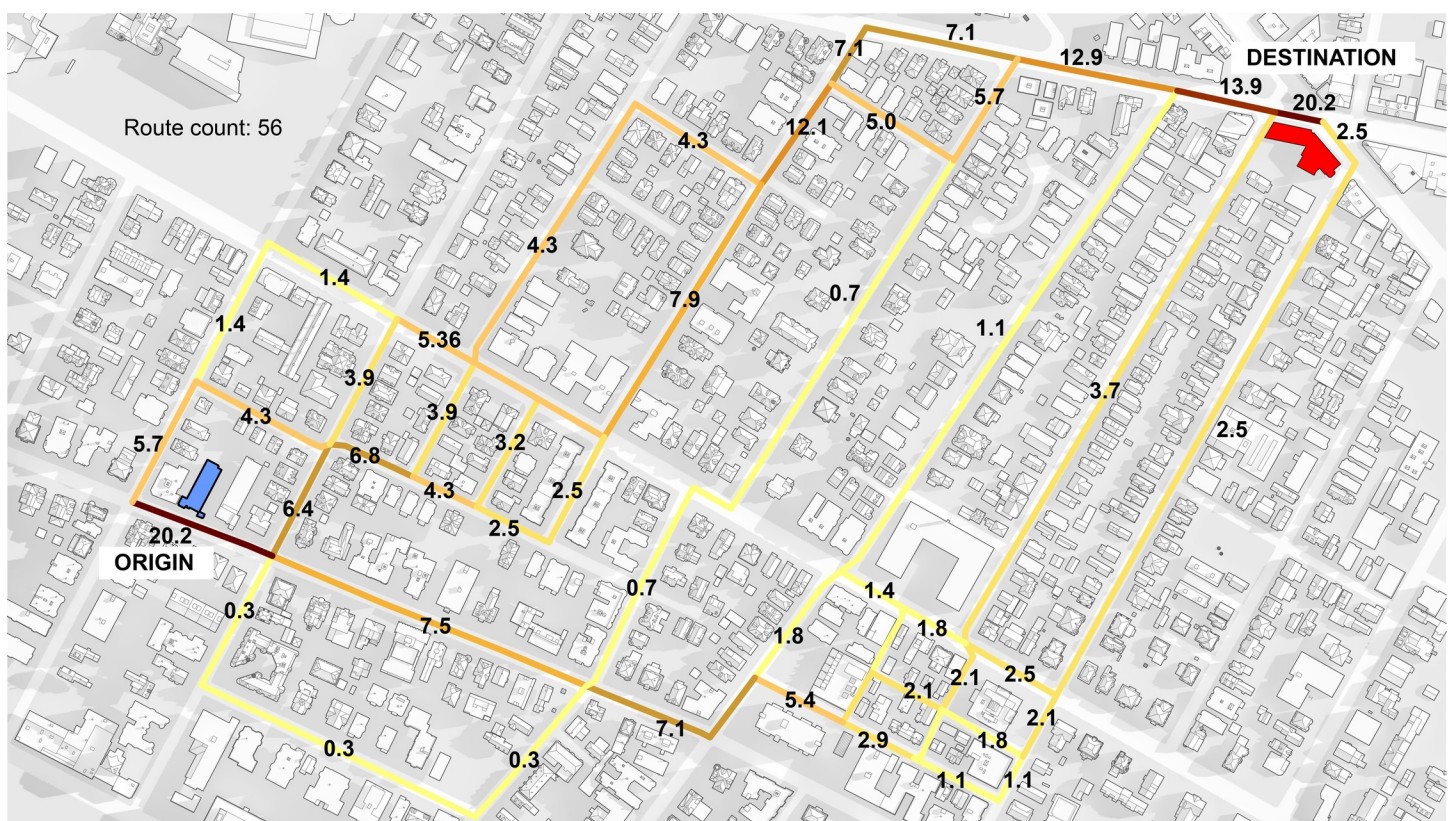

**Fig 2. Betweenness results for street segments for a single origin-destination pair, where the origin has a weight of "100" (of which only 20.2 are routed due to a distance decay effect) along all routes that are up to 15% longer than the shortest path.**

data on dominant trip generation types during morning and evening peak periods in the US context [58] and discussed dominant trip types with planners in Melbourne, which led to a selection of ten potentially important origin-destination pairs for our model, highlighted in gray in Fig 3. Patronage betweenness models were calculated using a maximum search radius of 800m, since the median pedestrian trip length in Melbourne's inner suburbs is 563m and

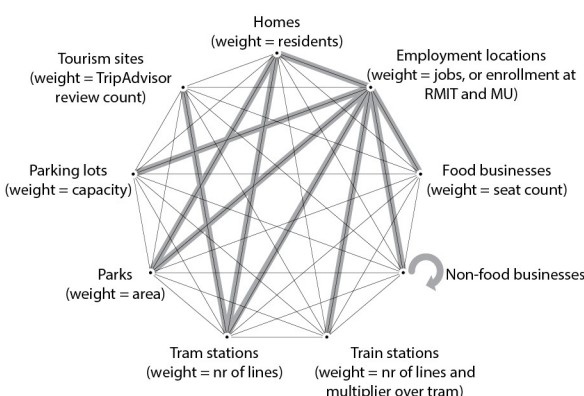

**Fig 3. Potential 36 origin-destination pairs for pedestrian trips in Melbourne.** Gray highlights include the ten O-D pairs for which trips were modeled. Weights describe which attribute was used as origin or destination weight in the analysis.

the average length is 770m [59]. We expected trips to be shorter in our CBD study area. We used a detour ratio of 1.15, and a distance-decay coefficient of 0.002 (in meters) for these ten trip types (S5 Table in S1 File). The two exceptions were flows between train stations and employment locations, which were calculated with a beta value of 0.001 assuming a greater propensity for regional commuters to walk further to/from train stations, and parking lot-employment flows which were calculated within a 400m radius, assuming that drivers are not willing to park more than a 5-minute walk from their destination. The model produced predictions of pedestrian flow values for all street segments in the city, including the locations used by the pedestrian counting sensors.

## Statistical methods for model calibration

We first tested how each of these estimated flows correlate with sensor counts during AM, lunch and PM peak periods and only included those flow types in a given year's models that exhibited 5% or greater positive bivariate correlations with observed counts in the same year. As a result, the list of included flows can vary for AM, Lunch, and PM models and may change from year to year (S1 Table in S1 File). Any negative correlations were not included in the model. Even though our exploratory analysis suggested that negative effects could increase the model fit, we saw no clear theoretical basis for their inclusion. In case of land use development, a negative coefficient may signal that adding more of a certain kind of land use can produce a modal shift away from walking (e.g., a car-oriented mall or single-family homes). But the variables in this model are not land use quantities but estimated pedestrian flows—we predict higher flows when land uses are closer or more favorably configured so as to generate more pedestrian trips, not car trips.

In order to calibrate trip-generation rates for each type of O-D flow, we regressed the selected pedestrian flow types against pedestrian volumes recorded by Melbourne's automated Pedestrian Counting System at the same locations [60, 61]. Pedestrian flows modeled on a given year's CLUE data constitute the key independent variables, which are calibrated on observed pedestrian counts recorded by sensors in June—roughly middle of the year. For instance, predicted flows modeled on 2013 CLUE data (which include built environment updates completed before January 2014) were regressed against June 2014 count data. Only business days were included to calibrate weekday AM peak, lunch peak, and PM peak models. S5 Table in S1 File provides descriptive statistics for the independent variables in 2014. The (at most ten) pedestrian flow estimates were combined with temperature and rain records for corresponding time periods.

Given that the pedestrian flow estimates are unique to each sensor, while the weather and day-of-week indicators vary jointly for all sensors, an ordinary least squares estimation was deemed unsuitable and we instead used a multi-level model (MLM). The multi-level model was specified to have fixed slopes and varying intercepts—the effects of the pedestrian flows on sensor counts are constant across sensors, and random effects were included to capture the impacts of unobserved variables for each sensor as well as day-of-week.

There is considerable collinearity among different pedestrian flow types, since a number of them (e.g., walks from homes to jobs, walks from train stations to jobs) originate from or head towards the same locations over a limited street network. Collinearity can violate regression assumptions and lead to biased coefficient estimates, including multi-level models. To address this, we first eliminated collinear variables and negative coefficients from the multi-level model to produce a more parsimonious model. However, since collinearity between different types of pedestrian flows is unavoidable (violating regression assumptions), a more flexible approach was needed. We therefore also calibrated the model on five different machine

learning specifications, which are designed to account for collinearity as well as non-linear relationships between variables that some previous studies have emphasized [62, 63]. These include Stochastic Gradient Descent (SGD), Support Vector Regression (SVR), Random Forest (RF), Bootstrap Aggregation (BAG), Gradient Boosting (GB), and Gaussian Process (GP). In order to address overfitting concerns related to machine learning models, we employed 10-fold cross-validation to randomly create training (80%) and validation (20%) datasets from the overall training dataset with replacement and shuffling [66]. Optimal hyperparameters were chosen such that the validation score was maximized, and were subsequently used for prediction using the test dataset. Each model's predictive accuracy was compared against observed pedestrian counts using metrics such as R-squared ($R^2$), mean absolute error (MAE), and root mean squared error (RMSE). Higher predictive accuracy is indicated by larger $R^2$ and smaller MAE and RMSE values.

The left-side of Table 3 reports estimation results for the MLM and the ultimately chosen SVR model calibrated on June 2014 counts, with sensor-level dummy variables (S6 Table in S1 File illustrates results from other machine learning models we tested). While the multi-level model (see also S3 Table in S1 File) achieves a good fit in Table 3 (AM period $R^2$ 0.92, lunch period $R^2$ 0.96, PM period $R^2$ 0.93), each of the machine learning techniques we tested outperformed the multi-level regression (S6 Table in S1 File). The random forest (RF) approach produced the highest fit on calibration data (AM period $R^2$ 0.99, lunch period $R^2$ 0.99, PM period $R^2$ 0.99). However, achieving a good fit to the calibration data is necessary but not sufficient to produce reasonably accurate predictions.

We subsequently tested how accurately each calibrated model can predict pedestrian activity at the same sensor locations on out-of-sample data for the same month a year later (June 2015) by changing the built environment data in the model to reflect 2015 conditions, and assuming pedestrian flow coefficients do not change. The accuracy of these predictions is reported on the right side of Table 3 and in S6 Table in S1 File. Here we see that the MLM model falls behind machine learning results, of which the SVR and BAG approaches achieved the best predictive fit. While lower fit values were expected for out-of-sample predictions, we still find relatively robust predictive accuracy a year after calibration (SVR AM period $R^2$ 0.78, lunch period $R^2$ 0.82, PM period $R^2$ 0.74). The RF approach, which performed best on calibration data, did not achieve as high prediction accuracy on out-of-sample data. A well-known shortcoming of Random Forest models is the tendency to overfit to training data [64, 65]. Since bootstrap aggregation using SVR (i.e., the BAG model) did not provide significant improvements in prediction accuracy over SVR, we chose SVR as the most reliable model for

**Table 3. Goodness of fit results of MLM and the chosen SVR model for 10 types of pedestrian flows combined with weather and day-of-week variables, with sensor-level dummies.** Left: calibration results on June 2014 data. Right: prediction results on June 2015 data.

| Calibration on June 2014 data (Mon-Fri) | | | | Prediction on June 2015 data (Mon-Fri) | | | |
|---|---|---|---|---|---|---|---|
| Peak | Metric | MLM | SVR | Peak | Metric | MLM | SVR |
| AM | R2 | 0.92 | 0.92 | AM | R2 | 0.74 | 0.78 |
| | MAE | 0.13 | 0.16 | | MAE | 0.35 | 0.33 |
| | RMSE | 0.31 | 0.31 | | RMSE | 0.62 | 0.56 |
| LUNCH | R2 | 0.96 | 0.96 | LUNCH | R2 | 0.75 | 0.82 |
| | MAE | 0.11 | 0.15 | | MAE | 0.36 | 0.29 |
| | RMSE | 0.18 | 0.19 | | RMSE | 0.49 | 0.42 |
| PM | R2 | 0.93 | 0.93 | PM | R2 | 0.71 | 0.74 |
| | MAE | 0.15 | 0.16 | | MAE | 0.42 | 0.37 |
| | RMSE | 0.28 | 0.27 | | RMSE | 0.68 | 0.63 |

subsequent analyses as it converges faster than BAG. S2 Fig in S1 File illustrates functional relationship between our predictors and outcomes in the SVR model, confirming non-linearity among some of them.

Note that the results in Table 3 are estimated by including sensor-level dummy variables, which helps improve the model predictions at sensor locations, but these dummy variables (capturing unobserved effects) cannot be used on other streets that lack sensor data. Similar model fits without sensor dummies are presented in S2 Table in S1 File.

The model is designed to capture urban form and land-use accessibility effects on walking trips. It also includes behavioral variables that adjust trip distances, decay rates, as well as assumptions about route choice. However, we acknowledge that the model is not a complete representation of all factors shaping walking behavior. It does not account for the social characteristics of streets, cultural factors, differences in individual needs and abilities, or variable sidewalk qualities on different streets, which have also been shown to impact walking behavior [66].

## Results

One of the key objectives of this study was to understand what might explain year-on-year changes in pedestrian activity on city streets. Pedestrian flows illustrate complex dynamics that are shaped by individual preferences, demographic biases (e.g., age, gender, and income effects), social norms (e.g., walking culture in a city or the lack thereof), weather conditions, time of day, and land-use interactions in the built environment. We were particularly interested in exploring how a pedestrian flow model calibrated on observed counts from a particular year can be used to predict changes in pedestrian activity that result from changes in the built environment—land use changes, including updated residential, job, and amenity locations—in subsequent years. Towards this end, we test four questions in the following subsections. (a) Does including land-use interactions improve model prediction accuracy? (b) Which pedestrian flow types are dominant during peak travel periods? (c) Does the accuracy of model predictions remain stable over time? And (d), does the relative importance of different pedestrian flows remain stable over time?

### Does including land-use interactions improve model prediction accuracy?

To address this question, we used the SVR model calibrated on June 2014 data and examined how the prediction accuracy changes when pedestrian flow variables between origin-destination land uses are removed from the model (i.e., the explanatory variables are reduced). We implemented this with two separate setups. The first setup (Table 4 left) includes dummies, which account for unobserved effects at the sensor level. Even when built-environment measures (i.e., pedestrian flows) are removed from the set of explanatory variables on which the model is calibrated, these dummy variables help distinguish pedestrian count variations unique to each sensor, which is at least in part due to differences in built environment conditions around each sensor. We find that adding pedestrian flows as predictors to the model increases $R^2$ values by approximately 3% during the morning peak hour, 1% at lunch, and 3% during the evening peak hour. A notably larger decrease in error, and thus a better fit, is also demonstrated through MAE and RMSE values. This type of model may be useful when predicting future pedestrian flows on streets where counters have been in place during prior periods for calibration. The dummy variables can absorb sensor-specific effects that may result from omitted variables, yielding higher accuracy and more reliable coefficient estimates. However, the dummy variables cannot be used when we want to predict changes in pedestrian activity on streets without prior pedestrian count data—on neighboring streets where sensors are not available.

The right-hand side of Table 4 thus presents the same model without sensor-level dummies. This model can be used to forecast pedestrian activity on any street based on nearby sensor

**Table 4. Evaluating impact of built-environment measures on prediction with sensor dummies (left) and without (right).** Prediction metrics for 2015 using a model calibrated on 2014 are presented.

| Peak | Metric | With sensor dummies | | | Without sensor dummies | | |
|---|---|---|---|---|---|---|---|
| | | SVR (w/o BE) | SVR (with BE) | Change (%) | SVR (w/o BE) | SVR (with BE) | Change (%) |
| AM | R2 | 0.76 | 0.78 | +3% | 0.07 | 0.53 | +657% |
| | MAE | 0.37 | 0.33 | -11% | 0.79 | 0.51 | -35% |
| | RMSE | 0.58 | 0.56 | -3% | 1.2 | 0.8 | -33% |
| LUNCH | R2 | 0.81 | 0.82 | +1% | 0.08 | 0.57 | +613% |
| | MAE | 0.28 | 0.29 | +4% | 0.77 | 0.46 | -40% |
| | RMSE | 0.43 | 0.42 | -2% | 1.11 | 0.7 | -37% |
| PM | R2 | 0.72 | 0.74 | +3% | 0.06 | 0.55 | +817% |
| | MAE | 0.4 | 0.37 | -8% | 1.05 | 0.59 | -44% |
| | RMSE | 0.66 | 0.63 | -5% | 1.39 | 0.9 | -35% |

calibration from prior years. When the pedestrian flow variables are eliminated, the SVR model only contains weather, time, and day-of-week information, resulting in identical predictions for all sensors in the city for each time period. As expected, leaving out dummies substantially increases the contribution of land-use based pedestrian flow variables. The SVR predictions without pedestrian flows in the model has an $R^2$ of 7% in the morning, 8% at lunch, and 6% in the evening. It increases to 53% in the morning, 57% at lunch, and 55% in the evening when pedestrian flows between land uses are added to the model. S4 Table in S1 File additionally shows a "null hypothesis" test, confirming that a model with updated built environment flows produces a better fit to observations than one where built environment flows are assumed to remain constant at the previous year levels. We thus conclude that pedestrian flow estimates derived from land-use interactions in the built environment substantially improve the model's accuracy, increasing the $R^2$ values by at least six times. A model with pedestrian flow coefficients calibrated on a prior year can be used to forecast changes in pedestrian activity that result from new development projects with a precision range of 50–60% in terms of $R^2$ values.

Fig 4 illustrates the absolute predicted AM peak pedestrian flows for an average workday in June 2015, based on June 2014 calibration. Pedestrian flow predictions are obtained for all sidewalk segments and crossings, not only those fronting the sensors. The results indicate flows up to around 4,000 pax/hr in dense commercial areas and close to major train stations, similar to observed calibration counts in such areas.

Fig 5 shows the predicted change in foot-traffic in 2015, due to the built environment transformations that took place between 2014 and 2015. While most predicted changes are positive as expected—Melbourne is a growing city—some streets are also predicted to lose foot-traffic due to a decrease in residents or jobs around them, or due to the development of new destinations in their vicinity, which can shift destination choices and trip distributions.

## Which pedestrian flow types are dominant during peak travel periods?

It is also important to examine which particular pedestrian origin-destination flows explain the foot-traffic counts observed in the morning, lunch, and evening peak hours from a planning perspective. We calculated the feature importance of pedestrian flows included in the model, defined as the decrease in model fit when values of that attribute are randomly shuffled. This is a model-agnostic computational approach, and can be interpreted to be indicative of the extent to which the model depends on that particular attribute [65, 66]. Fig 6 shows the contribution of each pedestrian flow type on the models' predictive accuracy (as measured by $R^2$). The axes inside boxes indicate median contributions towards $R^2$, the boxes extend from

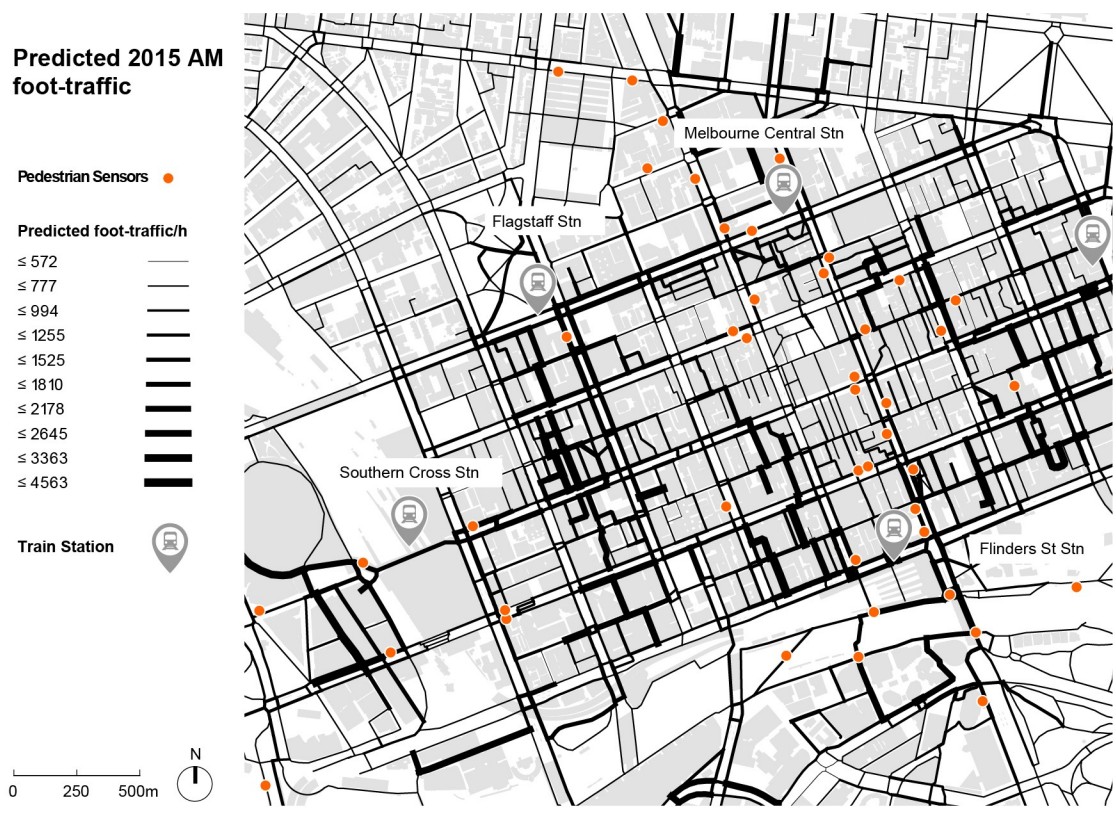

**Fig 4. Predicted pedestrian flows during a typical workday AM peak period (8.00–9.00AM) in June 2015, based on June 2014 calibration.**

the 25th to the 75th percentile ranges, while the length of the whiskers is 1.5 times the interquartile range and outliers are denoted by dots.

The predicted weekday morning peak (8-9AM) foot-traffic is dominated by pedestrian trips between train stations and employment locations, which contribute roughly 60% to the total $R^2$ outcome of 53%. There were around 326,000 people who commuted to work using public transport (train, bus, tram or ferry) in Greater Melbourne in 2016, of whom 75% took the train [67]. There are six heavy-rail stations throughout Melbourne, of which the Southern Cross station is the largest. Our model accounts for these size differences by weighting stations by the number of rail lines they serve.

Pedestrian trips between jobs and parking lots contribute 30% to the $R^2$ outcome, followed by trips between homes and jobs (23%). Flows between homes and parks, and jobs and parks contribute slightly over 10% each. Walks to and from tram stops contribute up to 5% to the overall model fit. Even though walking trips from homes to transit stations, transit stations to employment locations, and directly from homes to employments locations may illustrate different legs of trips headed to work in the morning peak period, we treat them separately here because our model does not seek to estimate individual agents' daily activities as complete trips, but rather pedestrian flows on individual street segments.

Lunch hour foot-traffic predictions are largely dominated by walking trips between amenities and other amenities, which contribute over 60% to the total $R^2$ outcome of 57%. This type of foot-traffic is also spatially correlated with flows from jobs to eateries, although the effect is notably lower. Other O-D trips are less pronounced during lunch, though walks between jobs and parking lots, and jobs and train stations contribute around 20% each.

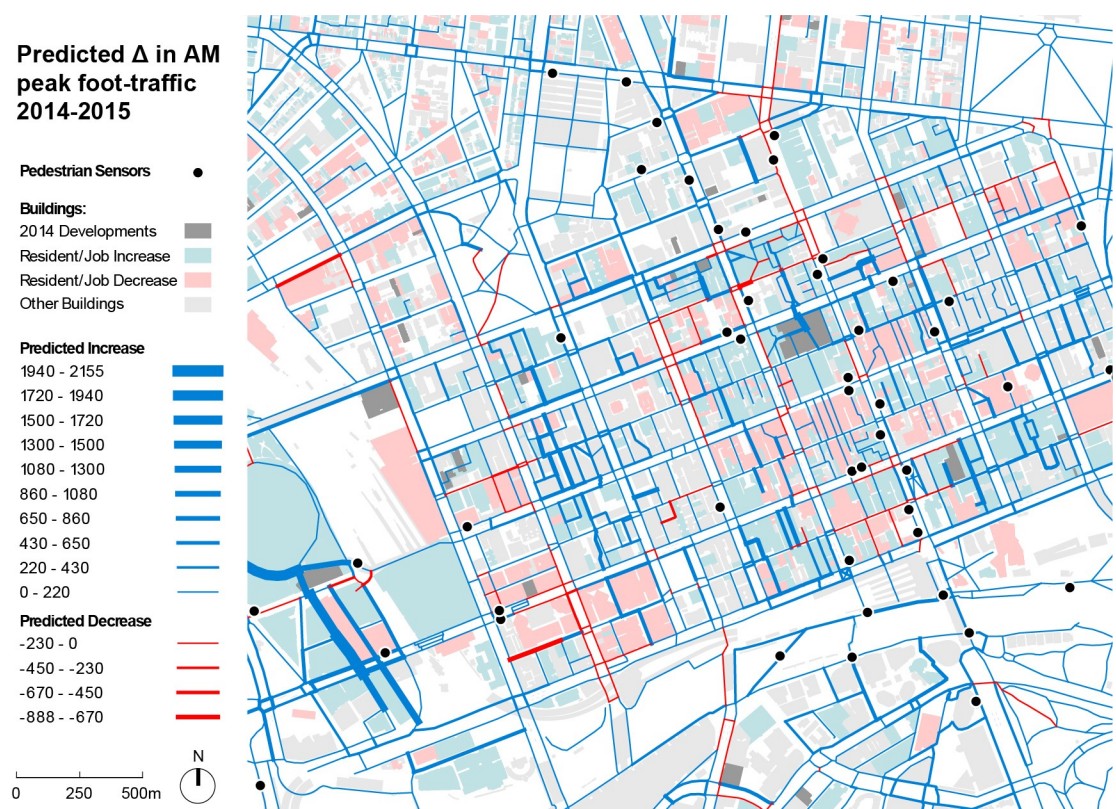

**Fig 5. Predicted change in pedestrian flows during a typical workday AM peak period (8.00–9.00AM) in between June 2014 and 2015.**

Evening peak hour predictions are explained by a more heterogeneous mix of trips. Walks between jobs and train stations dominate, similar to the morning peak hour, contributing around 40% to the total $R^2$ outcome of 59%. Pedestrian flows between jobs and tram stops, and jobs and homes each contribute around 20%. Walks between amenities and other amenities, as well as walks between parks and homes, and parks and jobs contribute to the PM peak model prediction accuracy by 5–10%.

Even though dominant flows were identified across all sensors for each of the three observation periods, pedestrian flows in neighborhoods that contain different land-use patterns and demographics should not be expected to feature the same dominant flows. For instance, lunch time trips between amenities are not going to dominate in residential neighborhoods that have no amenities, and trips between workplaces and train stations cannot dominate foot-traffic in an area without a train station. Neighborhoods covered by Melbourne's pedestrian sensors do vary in land-use characteristics and pedestrian flow types and the average hierarchy of model contributions across all sensors shown in Fig 6 should be interpreted with this caveat in mind. The lengths of the box-plots (e.g., the interquartile ranges) demonstrate this heterogeneity-driven variability, which is small in most cases but substantial in a few cases.

## Does the accuracy of model predictions remain stable over time?

We found that built environment changes explain around 50–60% of variation in observed foot traffic changes during the same month a year later. Such changes can include shifts in the spatial distribution of residences or jobs that result from new development activity, or the

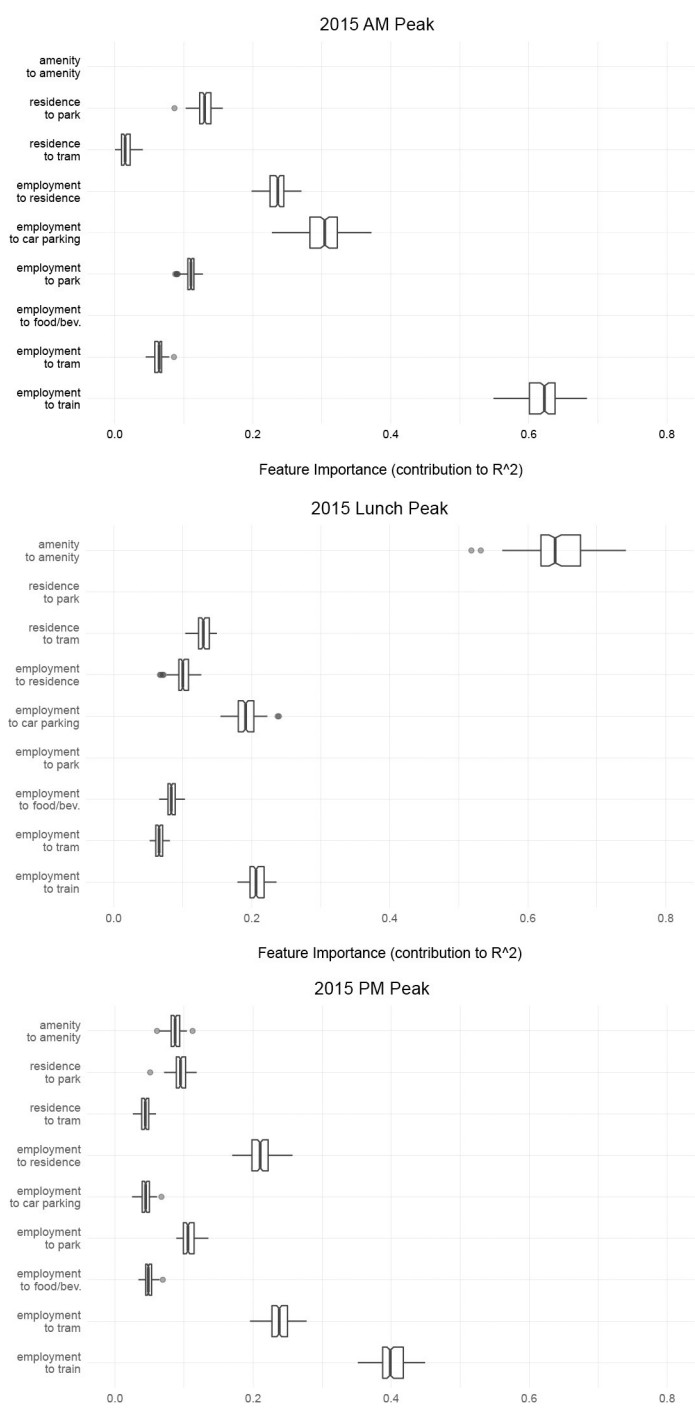

**Fig 6. Effect of pedestrian flow measures on model predictions without sensor-level dummies during AM peak, Lunch peak, and PM peak.**

formation of new food and beverage establishments and retail amenities, which are reflected through updates in the CLUE data. However, passing time can also bring about changes in pedestrian behavior, captured through model coefficients, raising a planning practice-relevant

question about how often a calibrated model requires an 'update'. How many years into the future can the model reliably forecast? How well does a model calibrated in 2014 predict pedestrian flows on updated CLUE data in 2016, 2017, 2018, or 2019? We expected the predictive accuracy to decline over time due to the possibility of new and omitted variables emerging over time (e.g., infrastructure changes not included in our model, demographic changes, the proliferation of new mobility services, etc.) or changes in pedestrian behavior captured through the calibrated coefficients (i.e., changes in destination, mode, or route preferences). To address this question, we calibrated a series of additional models on each year, where data was available and used each of these models to predict pedestrian activity with updated CLUE data in the following years up to 2019, the most recent data year that was available to us. Table 5 presents each of these models' RMSE values for predictions on gradually successive years. The smaller the RMSE values, the higher the predictive accuracy.

As already indicated in Table 4, an AM peak model calibrated in 2014 achieves an RMSE of 0.8 for predictions in 2015. However, its predictive accuracy declines with rising error values in 2016 (0.84), 2017 (0.8), 2018 (0.86) and 2019 (0.86). A similar pattern is observed on a model trained on 2015 data, whose RMSE values are lowest the year after and gradually rise in successive years. In four years, RMSE increases by 12% on average across AM, lunch, and PM peak models, compared to the baseline values from the first prediction year. In a three-year span, RMSE values rise by 14%, and in a two-year span by 12% on average. We also note a 7% change in a five-year span, which is likely smaller due to fewer (only three) pairwise comparisons that can be made with a five-year gap in Table 5. We thus conclude that using annually updated CLUE data enables a pre-calibrated model to predict changes in foot-traffic with about 12% loss in accuracy through four years following the first prediction year.

The findings in Table 5 suggest that the lunch period model remains most stable over time, where in a five-year span, errors increase by 7% on average, compared to the first year of prediction. The corresponding error increase for the AM period is 10% and for the PM period 12%. This may be due to shifting transportation policies that affect AM and PM travel (e.g., fare subsidies to incentivize off-peak travel on congested public transport), changes in work culture (e.g., work from home or staggered working hours) or changes in social culture (e.g., changing habits for eating out or shopping). The lunch period appears least affected by such year-on-year shifts.

**Table 5. Examining stability of model prediction performance without sensor-level dummies over time (using RMSE).**

| Peak | 2014 | 2015 | 2016 | 2017 | 2018 | 2019 |
|------|------|------|------|------|------|------|
| AM | TRAIN | 0.8 | 0.84 | 0.8 | 0.86 | 0.86 |
| | - | TRAIN | 0.61 | 0.62 | 0.8 | 0.76 |
| | - | - | TRAIN | 0.57 | 0.62 | 0.68 |
| | - | - | - | TRAIN | 0.78 | 0.85 |
| | - | - | - | - | TRAIN | 0.7 |
| LUNCH | TRAIN | 0.7 | 0.7 | 0.61 | 0.7 | 0.68 |
| | - | TRAIN | 0.53 | 0.73 | 0.67 | 0.68 |
| | - | - | TRAIN | 0.72 | 0.59 | 0.81 |
| | - | - | - | TRAIN | 0.68 | 0.65 |
| | - | - | - | - | TRAIN | 0.51 |
| PM | TRAIN | 0.9 | 0.91 | 0.8 | 0.94 | 0.99 |
| | - | TRAIN | 0.66 | 0.77 | 0.96 | 0.88 |
| | - | - | TRAIN | 0.76 | 0.61 | 0.83 |
| | - | - | - | TRAIN | 0.95 | 0.95 |
| | - | - | - | - | TRAIN | 0.79 |

Melbourne's unique capability to track pedestrian counts continuously makes it theoretically possible to recalibrate the model continuously with updated hourly data from the pedestrian counter API, potentially in an automated manner. However, since comprehensive CLUE data—which are necessary for accurate model calibration—are renewed annually, the minimal practical interval for recalibration is a year. If the model were to be used in an official capacity for "pedestrian impact assessments" of urban design or development plans, our results show that prediction coefficients would benefit from being recalibrated each year.

## Does the relative importance of different pedestrian flows remain stable over time?

We also examined whether the relationships between pedestrian flows and the built environment captured through the model coefficients remain stable over time, or whether pedestrians' behavioral responses to the built-environment modifications change over time due to adjustments in attitudes, preferences, or socio-cultural evolution [36]. We note that the calibrated coefficients could also change over time due to a number of conditions that remain unobserved in our model—demographic changes in the city, annual shifts in tourism, new policies that promote more walking, etc.

In order to explore the stability of the impacts of different pedestrian flows (i.e., the model coefficients), we recalibrated the model on only those sensors that were active throughout all six years (2014–2019) and eliminated those that were either added or removed throughout the years. This was necessary because if new sensors are added to a residential district in a particular year, while keeping other sensors constant, the source of variation in observed calibration data will shift and a different set of dominant O-D flows could be expected. By keeping the set of calibration sensors constant and evaluating O-D flows on the same exact streets and neighborhoods over time addresses this concern and focuses the analysis on behavioral change rather than differences in calibration areas. The plots in Fig 7 illustrate the importance of particular O-D flows on the model fit on the 25 sensors that were constant through calibration years 2014–2019.

In general, the rank order of calibration factors does not significantly shift over the years. During the AM peak, trips between train stations and workplaces dominate the flows on all years, followed by trips between parking lots and jobs, and homes and jobs. The relative importance of trips between parking lots and jobs notably decreases in 2015, and increases again in 2018. To make sense of this shift, we looked into Melbourne's transportation policies and discovered that in January 2015 Melbourne changed public transit pricing by dropping Zone 2 prices to the same level as Zone 1 (inner city). This appears to have increased train ridership in the AM peak, and may be a possible reason why we see a decrease in walking activity to parking lots [68]. However, the importance of walks between jobs and parking lots rebounded in 2018 in our model, when the city added more jobs than in other years (Table 1).

During the lunch hour, amenity-to-amenity trips dominate in each calibration year, but the relative contribution of these trips to the model fit appears to be slowly declining. Explaining this decline will have to remain the subject of future research, but we suspect it may be associated with the relative decrease in amenities per capita over the years. Table 1 shows that the number of amenities has not kept pace with the steady increase in the number of residents and jobs in the city center. While in 2014, there were 77 residents and jobs per each amenity in the study area, by 2018 the ratio had increased by almost 10% to 85 residents and jobs per each amenity. After 2016, the per capita provision of amenities was on a gradual rise again, possibly due to a lagged market response to the significant residential increase in the preceding years (Table 1).

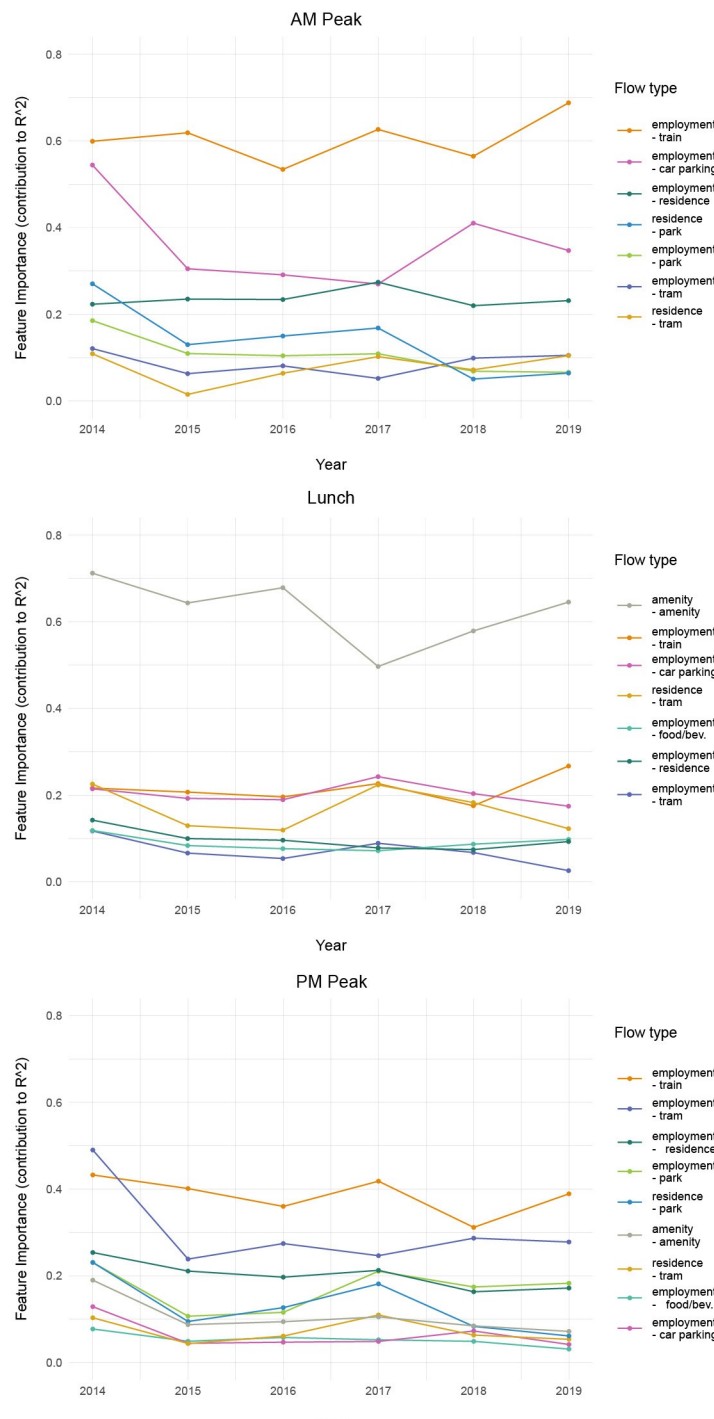

**Fig 7. Stability of feature importance from 2014–2019 for AM, lunch and PM peak periods (without sensor-level dummies).**

During the evening peak period, employment to train and employment to tram trips dominate, but overall, we find a more diverse and balanced set of flows in the PM period. Significant evening flows include walks to parks and green areas, which were also present in the AM model,

but not during lunch. Other PM flows include walks between tram stops and homes, amenities to amenities, and jobs to food and beverage establishments, contributing around 5–10% each.

## Discussion

Melbourne's population is growing rapidly. Partly testament to the high level of walkability and all-around quality of life that Melbourne offers, a 38.3% increase in daily population to the city is projected by 2030 [69]. A higher daytime population will result in more pedestrian activity and an increased need to analyze, design, plan, and manage the pedestrian realm.

The proposed model could serve at least two core functions in an official capacity. First, it could be used to illustrate pedestrian volumes on all city streets (not only those with sensors) at different time periods to produce a 'Pedestrian Census' of sorts. Analogous counts, showing average annual daily traffic (AADT) for vehicles have been available in US cities at the street segment-level for decades. Instead of having to count pedestrians on thousands of segments (as in AADT, which are typically based on automated vehicle counts), estimates can be generated based on calibration on far fewer counted segments. Such data could inform planners and transportation policy makers where new sidewalk improvements, landscaping investments, street furniture installations, and safe crossing interventions could be most beneficial. Unlike trend-analysis models [70], whose predictions are confined to sensor locations alone, our model can forecast location and time-specific pedestrian flows on all sidewalks and crosswalks within a reasonable range around calibration locations.

Second, the proposed model can be used to forecast how upcoming development proposals are likely to change pedestrian flows on surrounding city streets. These may include actual development proposals, new zoning propositions, or planning and urban design scenarios covering groups of parcels in some parts of the city. For instance, the model can be used in early phases of neighborhood redevelopment planning to assess how different urban design schemes would likely affect future pedestrian traffic, suggesting where commercial ground floor zoning is warranted, where public space investments are needed, etc. Alternatively, it could be used to examine proposed development effects on a specific, single lot—how a new office building is likely going to impact foot-traffic around it. Such use-cases could be achieved by hosting the calibrated model on an interactive, public web-map, enabling users to update land-use characteristics and built density variables on individual lots interactively to examine how proposed changes could impact foot-traffic on surrounding streets during different times of the day. The resulting Pedestrian Impact Assessment estimates can form a basis for seeking financial investments from developers to improve specifically affected pedestrian infrastructure—an already common practice for Traffic Impact Assessments. Unlike present Traffic Impact Assessments, which are typically performed by hired consultant teams separately for each project site, the online model could provide a widely accessible platform that developers, city officials, and interested community members could each use with transparency for better informed discussions concerning development impacts on non-motorized mobility. Such evidence could help equalize pedestrian-oriented policy and planning concerns with vehicle-oriented concerns.

The unique contributions of the model include (1) the capacity to estimate not only changing volumes of walking trips that result from new urban developments, but also their spatial distributions, (2) the ability to predict land-use induced changes in foot-traffic at a resolution of individual street segments with 50–60% accuracy within a year, and (3) the potential to evaluate how specific urban development projects could alter, and desirably benefit, foot-traffic in surrounding neighborhoods.

The model rests on a number of assumptions, which may be sensible but would benefit from further refinement. First, the model assumes that pedestrian counts at limited sensor

locations can be used to infer pedestrian flows between specific O-D pairs. However, if specific land use pairs are missing at available sensor locations (e.g., hospital to university), but available at other locations, then the model will be unable to capture and rank their importance. In planning new pedestrian sensor locations, it is therefore be important to include locations with as many heterogenous land-use pairs as possible. In future work, it also would be useful to analyze the dominant morning, lunch, and evening peak pedestrian flow patterns separately by type of neighborhood—majority residential, majority office, retail, industrial, or recreation districts. This would contribute to understanding how typical activity patterns vary by land-use mix and density in each type of neighborhood. However, detecting neighborhood-type specific dominant flows at different times of day would require a larger number of calibration sensors than what was available at present (there were 62 active sensors in Melbourne by the end of 2018).

Second, the model we presented generated pedestrian flows based on distance minimization and 15% detours over shortest paths. Pedestrian route choice literature has demonstrated that other factors, such as turns along the route, street crossings, elevation gain, or street characteristics passed along the way are also important factors for explaining observed path choice. Future work on path generation algorithms could incorporate more factors beyond distance or travel time, perhaps using weights that will have to be empirically tested, for route generation.

Third, the model presented here depends directly on land use interactions, but does not account for socio-economic differences between urban districts. Neighborhoods of different age groups, income, ethnicities, or cultures may exhibit different pedestrian trip behaviors that future research efforts could try to distinguish. We also did not factor in residential or commercial vacancies, and kept the transit network, pedestrian network, parks, tourist destinations, and university dorms constant from 2014–2019, even though some changes in their distributions may have occurred. Future research could further incorporate such variations as well. Despite these limitations, the relative stability of results on different years suggests that the model forecasts pedestrian activity in Melbourne with reasonable precision.

The month of June offered a convenient choice for model calibration, sampled approximately in the middle of the CLUE survey on built environment changes for a given year. This does not limit the model coefficients to June alone—the same coefficients can be used to forecast pedestrian flows during other months if combined with corresponding monthly dummy variables. However, months that have a distinctly different mobility pattern (e.g., the holiday break in December, or the hosting of the Australian Open tennis tournament in Melbourne in January) should have a separate model calibrated to identify their particularities. Separate models can be calibrated for weekend flows, holidays, or special events that take place in the city.

The analysis can be readily replicated in other cities that collect pedestrian counts and annual data on built environment changes. While the latter is rather common in cities in some industrialized countries, systematic collection of pedestrian counts still remains rare. Automated pedestrian counts have recently gained popularity in the U.S. context [70, 71]. While the model can also be calibrated with manually collected pedestrian counts, we hope this study can help encourage more cities to adopt automated and anonymous pedestrian counting systems that could lead to more evidence-based planning for non-motorized trips and street activities.

## Supporting information

**S1 Dataset. Data table used to estimate the models.**
(CSV)

**S1 File.**
(DOCX)

## Acknowledgments

We would like to thank the City of Melbourne for sharing data. We also thank urban designer Gerhana Waty from Hansen Partnership and Will Mcintosh and Alireza Pour from the City of Melbourne for sharing insights and background information about walking activity in the city, and the two anonymous reviewers for their comments and suggestions.

## Author Contributions

**Conceptualization:** Andres Sevtsuk.

**Data curation:** Andres Sevtsuk, Bahij Chancey.

**Formal analysis:** Andres Sevtsuk, Rounaq Basu, Bahij Chancey.

**Investigation:** Andres Sevtsuk.

**Methodology:** Andres Sevtsuk, Rounaq Basu, Bahij Chancey.

**Project administration:** Andres Sevtsuk.

**Supervision:** Andres Sevtsuk.

**Validation:** Rounaq Basu.

**Visualization:** Rounaq Basu, Bahij Chancey.

**Writing – original draft:** Andres Sevtsuk, Rounaq Basu, Bahij Chancey.

**Writing – review & editing:** Andres Sevtsuk, Rounaq Basu, Bahij Chancey.

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
