## [Decision Letter · Decision Letter 0]

18 Feb 2021

PONE-D-20-36700

We shape our buildings, but do they then shape us? A longitudinal analysis of pedestrian flows and development activity in Melbourne.

PLOS ONE

Dear Dr. Sevtsuk,

Thank you for submitting your manuscript to PLOS ONE. After careful consideration, we feel that it has merit but does not fully meet PLOS ONE’s publication criteria as it currently stands. Therefore, we invite you to submit a revised version of the manuscript that addresses the points raised during the review process.

We look forward to receiving your revised manuscript.

Kind regards,

Wenjia Zhang

Academic Editor

PLOS ONE

Journal Requirements:

4. We note that Figures 1 and 3 in your submission contain map images which may be copyrighted. All PLOS content is published under the Creative Commons Attribution License (CC BY 4.0), which means that the manuscript, images, and Supporting Information files will be freely available online, and any third party is permitted to access, download, copy, distribute, and use these materials in any way, even commercially, with proper attribution. For these reasons, we cannot publish previously copyrighted maps or satellite images created using proprietary data, such as Google software (Google Maps, Street View, and Earth). For more information, see our copyright guidelines: http://journals.plos.org/plosone/s/licenses-and-copyright.

4.1.    You may seek permission from the original copyright holder of Figures 1 and 3 to publish the content specifically under the CC BY 4.0 license. 

4.2.    If you are unable to obtain permission from the original copyright holder to publish these figures under the CC BY 4.0 license or if the copyright holder’s requirements are incompatible with the CC BY 4.0 license, please either i) remove the figure or ii) supply a replacement figure that complies with the CC BY 4.0 license. Please check copyright information on all replacement figures and update the figure caption with source information. If applicable, please specify in the figure caption text when a figure is similar but not identical to the original image and is therefore for illustrative purposes only.

Reviewers' comments:

Reviewer's Responses to Questions

**Comments to the Author**

1. Is the manuscript technically sound, and do the data support the conclusions?

Reviewer #1: Partly

Reviewer #2: No

2. Has the statistical analysis been performed appropriately and rigorously? 

Reviewer #1: No

Reviewer #2: No

3. Have the authors made all data underlying the findings in their manuscript fully available?

Reviewer #1: No

Reviewer #2: Yes

4. Is the manuscript presented in an intelligible fashion and written in standard English?

Reviewer #1: No

Reviewer #2: Yes

5. Review Comments to the Author

Reviewer #1: The article presents results of a statistical analysis of sensor data for counting the volume of pedestrian flow in Melbourne. The volume of pedestrian flows is correlated with factors of urban life (land use, …) to calibrate the model and test its predictive ability. The importance of trips by foot is growing and such data are rare but important for planning cities and supporting foot travel. Therefore, in principle, the reviewer supports the publication. However, the presentation of the results is not optimal and the statistical analysis and interpretation of the results also need improvement. The reviewer suggests a revision of the article along the following comments before publication.

1. The central criticism concerns the 11 factors (‘estimated pedestrian flow types: emp_trn, emp_trm, …) considered in the model for the pedestrian volumes. Figures 4 and 5 show that most of the factors seem to have little contribution, or not much importance. In the same time, it is obvious that the factors that have been taken into account cannot be complete. As the authors state in line 173 -176 many factors are not included. The reviewer is aware that here completeness cannot be achieved. However, it would be possible in the statistical analysis to consider that unknown factors (random effects) were neglected and how strong their influence is. It seems that random effects were considered by the authors, see line 196, but it is not explained in detail how these random effects influence the results and how important they are. Furthermore it is surprising that many of the factors (flow types) hardly make an explanatory contribution to the model. I think it would be appropriate to check whether the high complexity (high number of fixed variables) is justified or necessary (e.g. by using the Akaike Information Criterion).

2. More details should be provided in the presentation and discussion of the CLUE data used for the model calibration. In particular, information about the spatial resolution of the data would be important to ensure the reproducibility of the study and to be able to evaluate the results.

3. Line 33 – 38: One the one side the authors conclude that an update should take place every two years. One the other hand they found that structural changes influence pedestrian traffic. This seems strange. What if in two years no structural changes occur. Or if many structural changes happen in the period of one year?

4. Line 71 and other places in the manuscript. ‘Logitudinal’ is a term referring to spatial information. To my understanding the authors use it to describe a property in time. Is this intended?

5. Line 104 to 106: This filtering in the data seems important to me and I expect the results to be very sensitive. It should be investigated whether the chosen filtering is justified and how the results change when the filtering is changed.

6. Line 114:. What is meant with elastic? Larger variability? Of what quantity? More precision in the language is necessary.

7. Line 121: ‘… idiosyncratic …’ What is meant by the term? Can this be specified. E.g. Individual criteria of route choice, such as safety, comfort, …

8. Line 235 to 237: unprecise terminology: ‘… multiple forces …’ behavior or group dynamics is not a force.

9. Line 300: The study distinguishes between trips from home to work and trips from home to access public transit. But a trip from home to work can also include public transit. It is not clear to me how the model accounts for this duality.

10. Figure 4: The abbreviations used in the y-axis should be defined clearly

11. Table 3: What is the reason that the predictive power of the models change for AM and PM but less for the Lunchtime Periode?

Reviewer #2: Overall, the manuscript (titled: We shape our buildings, but do they then shape us? A longitudinal analysis of pedestrian flows and development activity in Melbourne) aims to make some contributions to forecasting of the effect of urban form, land uses, amenities and pedestrian walkway layout as well as climatic conditions changes over time using pedestrian flows. Three different time period are investigated based on updated fine grained built environment data available in Central Melbourne. The topic is appropriate for the journal PLOS ONE and will interest urban planner, urban designer, geographers, transportation planners, and policy makers. The manuscript is well-written; however, some issues remain to be addressed before the paper will be ready for publication.

It is known that pedestrian flows over time/space are well correlated. It would be good to have a sense of these correlations over time/space and to understand how the model does or not better than the null hypothesis i.e. pedestrian flows are staying the same/have very high correlation between the calibration year and the predicted year.

The paper does not make clear what/where is changing in the built environment how much is changing whether land use changes are marginal or not, how this affect or not pedestrian flow distribution or whether it is for example the pedestrian layout change or not and where during the calibration year and the predicted year and over the overall available period to have a sense of the magnitude of change.

With such detailed data set from the same reliable source, it would be good to know how the year-on-year means is changing or not and whether it is fully attributable to BE change or to exogenous factors. This is picked up in the last part of the paper – but not shown anywhere to begin with.

Please give the BEs and list of independent variables and their descriptive statistics. Whenever available the coefficients should also be given. Is this a ML black box model? The paper mention various weighting can these be described systematically.

Give indication of the pedestrian network area (cropped on figure 3) and of UNA running times.

Measuring overall model success with RMSE imply that scale information is preserved. RMSE does not necessarily increase with the variance of the errors. RMSE increases with the variance of the frequency distribution of error magnitudes. Does the model claim to measure overall demand? i.e. indirect linkage caused by external economic on both the overall levels of flow and the BE changes themselves? Dependencies might occur only directly. What is predicted need some clarification early on.

The comparison of the BE actual change versus the share of each source in the model would be of interest. This is key in a forecast model.

The model claims to combine the Huff model with Betweenness, could the authors give the formula used. This combination could give some rather non-standard elasticity effects unless done right.

In the city centre of Melbourne, it is surprising to see no relationship between non-food and food business in Figure 2, could this choice be explained/justified?

In the study area, there are well used publicly accessible indoor multi-level pedestrian path which are well connected with the outdoor network, could the authors clarify what is the extend of that network and in what way these omitted parts of the pedestrian network and flows affect the proposed model/results?

It is unclear why the model should be recalibrated every two years. Hopefully, the information requested above will also help articulate why it is so.

Figures 4 and 5 are unreadable (low resolution) and the variables are not defined/listed anywhere.

Please justify the specific radii size (800m). We don’t think that the US Transport Board reference is adequate when effectively the model claims for a context sensitive approach (Melbourne's travel surveys?).

There is still missing reference to layout change and its association with flow changes over time in this paper and software that provide the same.

What do the results tell us about the broader implications of pedestrian flow prediction?

A couple of points regarding ergonomic of the model – within a general design process of street improvement and change or monitoring where does the model stand?

Is it more likely to be used for exploring multiples options during early design exploration stage, or is it on a par with micro-simulation model?

In short, elaborate why the model or the tool is useful, and how the model (or tool) can be effectively used in the urban/transport planning and design practices, for example to appraise layout change (e.g. before and after Stanton St. or Bourke St. redevelopment, management change for example conversion to walking street (7 between 2014-17 – Melbourne walking plan) vs land use or amenities changes.

In the current state of the paper, it remains difficult to interpret results and value them, especially deviations to counts and the absence of level of service. The paper would increase its relevance when comparing results with another model (beside null model). For comparison, in theory there are multi-modal models (and therefore pedestrian models) available, including flow forecast, would it be possible to add a variation / modification of provided model, with corresponding results. Given the data set required it remains unclear if all components and considered methods are necessary as a whole for pedestrian forecast and thus if the model is really transferable as most cities in the world do not have such comprehensive data set.

6. PLOS authors have the option to publish the peer review history of their article (what does this mean?). If published, this will include your full peer review and any attached files.

Reviewer #1: No

Reviewer #2: No

---

## [Author Response · Author response to Decision Letter 0]

21 Apr 2021

Pls see the attached document with reviewer responses.

---

## [Decision Letter · Decision Letter 1]

14 Jun 2021

PONE-D-20-36700R1

We shape our buildings, but do they then shape us? A longitudinal analysis of pedestrian flows and development activity in Melbourne.

PLOS ONE

Dear Dr. Sevtsuk,

Thank you for submitting your manuscript to PLOS ONE. After careful consideration, we feel that it has merit but does not fully meet PLOS ONE’s publication criteria as it currently stands. Therefore, we invite you to submit a revised version of the manuscript that addresses the points raised during the review process.

We look forward to receiving your revised manuscript.

Kind regards,

Wenjia Zhang

Academic Editor

PLOS ONE

Reviewers' comments:

Reviewer's Responses to Questions

**Comments to the Author**

1. If the authors have adequately addressed your comments raised in a previous round of review and you feel that this manuscript is now acceptable for publication, you may indicate that here to bypass the “Comments to the Author” section, enter your conflict of interest statement in the “Confidential to Editor” section, and submit your "Accept" recommendation.

Reviewer #1: All comments have been addressed

Reviewer #2: (No Response)

2. Is the manuscript technically sound, and do the data support the conclusions?

Reviewer #1: Yes

Reviewer #2: Partly

3. Has the statistical analysis been performed appropriately and rigorously? 

Reviewer #1: Yes

Reviewer #2: I Don't Know

4. Have the authors made all data underlying the findings in their manuscript fully available?

Reviewer #1: Yes

Reviewer #2: Yes

5. Is the manuscript presented in an intelligible fashion and written in standard English?

Reviewer #1: Yes

Reviewer #2: Yes

6. Review Comments to the Author

Reviewer #1: The comments were essentially taken into account by the authors. Here are just two final feedbacks on the responses to the reviewer's comments:

4. Line 71 and other places in the manuscript. ‘Longitudinal’ is a term referring to spatial

information. To my understanding the authors use it to describe a property in time. Is this intended?

Answ:Both the CLUE data and the pedestrian count data are collected over time. The term

“longitudinal” is used in the paper to refer to temporal changes in pedestrian flows and the

built environment.

Repl to the Answ: This is no answer to my point. ‘Longitudinal’ is a term referring to spatial information (NOT temporal) and thus not used properly in the manuscript.

6. Line 114:. What is meant with elastic? Larger variability? Of what quantity? More precision in

the language is necessary.

Answ: In addition to using the Huff probability model to allocate trips to competing destinations

k, we include a Gravity Accessibility term, which ensures that the number of trips from [j]

to [k] also depend on the wight and proximity of destination k:

[, ] = [] ∙ [, ] ∙

[]

∙[,]

Consider for example a single trip origin (building) with 10 people in it, and a single

destination within its walking radius, such as a restaurant with a size of 1,000 sqft.

Depending on how far and big the destination is, we find the rate of trips sent to this

destination. Now, if another business establishment is added to the destination building

(adding a weight of 1,000 sqft), then that building obtains a new, larger weight (2,000sqft).

This would result in allocating more trips from the same origin, even when origin weight

stays the same at 10 people. Our elasticity effect ensures that when destination weights

expand, or when they are closer to the origin, trip generation increases.

Repl to the Answ: Many thanks for the explanation how the term ‘elastic’ should be understood. Please reconsider whether this explanation is of interest for potential readers, too! Or better choose a more appropriate term. Would ‘variabality’ of the model wrong?

Reviewer #2: Thank you for addressing most of the comments and criticisms

Three major criticism remains

Two are about the selection of the factors considered in the model:

1. the omission of the role of the pedestrian outdoor pathway layout in pedestrian route choice assignment: Shatu, et al. (2019) in a small study in Brisbane, Australia (N= 178) have shown that pedestrian route choice is mostly an angular distance minimization and secondarily Euclidean minimization. The pedestrian model proposed in Melbourne should at least include a route choice assignment that is aligned with this recent study of pedestrian route choice in Brisbane. Moreover, a large-scale study (Bongiorno, et al., 2021) using phone traces in the US (Boston and San Francisco, N= 552,478) also shows that pedestrian route choice is “vector-based” i.e., mainly angular “a cost that depends on the angular deviation of the street segment from the destination” and that there is asymmetry when OD are swapped. The paper shows that including this angular approach to pedestrian route choice increase model predictive power by 35%.

The route choice assignment following shortest angular path that would include 15% detour would be very different than shortest Euclidean path that include 15% detour used in the proposed model.

Given that UNA the software used in the paper includes directness metrics we expect that this pedestrian route choice preference and pedestrian outdoor pathway layout factor be included in the model and compared with the current route choice assignment based on Euclidean distance.

2. The pedestrian flow model proposed is longitudinal, yet it seems that the pedestrian outdoor pathway network remains the same from the base model and for the forecasted years. Could the authors provide information about the pedestrian outdoor pathway network layout between the baseline and the successive forecasted years: as to whether pedestrian outdoor pathway network layout changed and by how much/where? Whether if pedestrian outdoor pathway network layout changed the modelling assumed that it was the same or not? What are the possible consequence on the modelling?

3. Thank you for providing and extensive specification of the betweenness index and the definition of elasticity which is to say the least is unorthodox: e.g., is it possible that with an alpha = 1 then increasing a destination weight can potentially deprive other destinations of trips, which shouldn’t happen in a fully elastic model.

It seems that the betweenness index is the same that was used in Sevtsuk A., 2021, JAPA. It was then stated that “the model does not presently incorporate destination count elasticity, which should be addressed in future work.” Is this issue addressed in this paper? It should.

If not, the same disclaimer should be included, and the originality contribution of this paper would be somehow limited to the longitudinal modelling in Melbourne.

7. PLOS authors have the option to publish the peer review history of their article (what does this mean?). If published, this will include your full peer review and any attached files.

Reviewer #1: No

Reviewer #2: No

---

## [Author Response · Author response to Decision Letter 1]

21 Jun 2021

pls see attached response letter.

---

## [Decision Letter · Decision Letter 2]

7 Aug 2021

PONE-D-20-36700R2

We shape our buildings, but do they then shape us? A longitudinal analysis of pedestrian flows and development activity in Melbourne.

PLOS ONE

Dear Dr. Sevtsuk,

Thank you for submitting your manuscript to PLOS ONE. After careful consideration, we feel that it has merit but does not fully meet PLOS ONE’s publication criteria as it currently stands. Therefore, we invite you to submit a revised version of the manuscript that addresses the points raised during the review process.

We look forward to receiving your revised manuscript.

Kind regards,

Wenjia Zhang

Academic Editor

PLOS ONE

Journal Requirements:

Additional Editor Comments (if provided):

Because two reviewers provided positive comments, I provide a suggestion of minor revision. But I am confused by several issues when I read the paper. Please carefully address the following issues. If the revision is still unsatisfactory, I might need to find additional reviewers for another round of review.

First, in the "statistical methods" section, what are your dependent variables? & independent variables? How are they defined and measured? Are they street-link based, or sensor-based? How many samples in your models? Also, a descriptive statistics table should be provided. It is hard to judge the soundness based on your current description on methods, let alone to replicate your methods.

Second, as the built environment is a key word of your study, you need to provide a theoretical debate on why and how selecting such built environment variables, and what are their associations with walking behaviors? In this case, a literature review is needed to add in the Introduction or variable section. There are lots of related studies; you may refer to Handy, 2018, Enough with the “ D ’ s ” already—Let ’ s get back to “ A .” and Zhang et al., 2020, Nonlinear effect of accessibility on car ownership in Beijing: Pedestrian-scale neighborhood planning.

Third, the modeling process reported in Pages 17-26 are difficult to follow. I would suggest creating a table to show your modeling settings one by one in advance (e.g., in the method section), instead of reporting your modeling processes in the result section.

Fourth, why including so many types of machine-learning methods but only selecting one or two to report the results? What are the standard of your modeling selections? Also, you mentioned the non-linearity as an advantage of machine-learning, but your study does not show any nonlinear analysis. It seems you mainly used the models for prediction, not about the relationships between variables, not about nonlinearity.

Reviewers' comments:

Reviewer's Responses to Questions

**Comments to the Author**

1. If the authors have adequately addressed your comments raised in a previous round of review and you feel that this manuscript is now acceptable for publication, you may indicate that here to bypass the “Comments to the Author” section, enter your conflict of interest statement in the “Confidential to Editor” section, and submit your "Accept" recommendation.

Reviewer #1: All comments have been addressed

Reviewer #2: All comments have been addressed

2. Is the manuscript technically sound, and do the data support the conclusions?

Reviewer #1: Yes

Reviewer #2: Yes

3. Has the statistical analysis been performed appropriately and rigorously? 

Reviewer #1: Yes

Reviewer #2: Yes

4. Have the authors made all data underlying the findings in their manuscript fully available?

Reviewer #1: Yes

Reviewer #2: Yes

5. Is the manuscript presented in an intelligible fashion and written in standard English?

Reviewer #1: Yes

Reviewer #2: Yes

6. Review Comments to the Author

Reviewer #1: (No Response)

Reviewer #2: (No Response)

7. PLOS authors have the option to publish the peer review history of their article (what does this mean?). If published, this will include your full peer review and any attached files.

Reviewer #1: No

Reviewer #2: No

---

## [Author Response · Author response to Decision Letter 2]

26 Aug 2021

Pls see attached response letter

---

## [Editor Report · Decision Letter 3]

6 Sep 2021

We shape our buildings, but do they then shape us? A longitudinal analysis of pedestrian flows and development activity in Melbourne.

PONE-D-20-36700R3

Dear Dr. Sevtsuk,

We’re pleased to inform you that your manuscript has been judged scientifically suitable for publication and will be formally accepted for publication once it meets all outstanding technical requirements.

Kind regards,

Wenjia Zhang

Academic Editor

PLOS ONE